# Molecular Typing, Characterization of Antimicrobial Resistance, Virulence Profiling and Analysis of Whole-Genome Sequence of Clinical *Klebsiella pneumoniae* Isolates

**DOI:** 10.3390/antibiotics9050261

**Published:** 2020-05-17

**Authors:** Andrey Shelenkov, Yulia Mikhaylova, Yuri Yanushevich, Andrei Samoilov, Lyudmila Petrova, Valeria Fomina, Vitaly Gusarov, Mikhail Zamyatin, Dmitriy Shagin, Vasiliy Akimkin

**Affiliations:** 1Central Research Institute of Epidemiology, Novogireevskaya str., 3a, 111123 Moscow, Russia; mihailova@cmd.su (Y.M.); yanushevich@cmd.su (Y.Y.); samojlov@cmd.su (A.S.); shagdim777@gmail.com (D.S.); vgakimkin@yandex.ru (V.A.); 2National Medical and Surgical Center named after N.I. Pirogov, Nizhnyaya Pervomayskaya str., 70, 105203 Moscow, Russia; lutix85@yandex.ru (L.P.); med_2006@mail.ru (V.F.); gusarov1974@mail.ru (V.G.); mnz1@yandex.ru (M.Z.); 3Pirogov Russian National Research Medical University, Ostrovitianova str., 1, 117997 Moscow, Russia

**Keywords:** Enterobacteriaceae, antibiotic resistance, whole-genome sequencing, genomic epidemiology

## Abstract

*Klebsiella pneumoniae* is one of the most important pathogens concerned with multidrug resistance in healthcare-associated infections. The treating of infections caused by this bacterium is complicated due to the emergence and rapid spreading of carbapenem-resistant strains, which are associated with high mortality rates. Recently, several hypervirulent and carbapenemase-producing isolates were reported that make the situation even more complicated. In order to better understand the resistance and virulence mechanisms, and, in turn, to develop effective treatment strategies for the infections caused by multidrug-resistant *K. pneumoniae*, more comprehensive genomic and phenotypic data are required. Here, we present the first detailed molecular epidemiology report based on second and third generation (long-read) sequencing for the clinical isolates of *K. pneumoniae* in the Russian Federation. The data include three schemes of molecular typing, phenotypic and genotypic antibiotic resistance determination, as well as the virulence and plasmid profiling for 36 *K. pneumoniae* isolates. We have revealed 2 new multilocus sequence typing (MLST)-based sequence types, 32 multidrug-resistant (MDR) isolates and 5 colistin-resistant isolates in our samples. Three MDR isolates belonged to a very rare ST377 type. The whole genome sequences and additional data obtained will greatly facilitate further investigations in the field of antimicrobial resistance studies.

## 1. Introduction

*Klebsiella pneumoniae* is a Gram-negative, encapsulated, non-motile bacterium, being one of the six most important healthcare-associated pathogens (ESKAPE) which often exhibit multidrug resistance and/or are highly virulent, thus causing major healthcare concerns. The virulence of *K. pneumoniae* is determined by numerous factors including the polysaccharide capsule that allows the bacterium to evade phagocytosis by the host [1], type 1 and 3 fimbriae responsible for bacterial cell adhesion [1] and various siderophores used for iron uptake [2]. Several hypervirulent types of *K. pneumoniae* carry genes associated with life-threating invasive diseases [3]. 

The treating of infections associated with this bacterium is also complicated due to the emergence of carbapenem-resistant strains revealed in the mid-2000 s. This infection has spread around the world, and it represents a serious problem due to its high mortality rate, which can sometimes exceed 50% [4]. Making matters worse, recently an isolate was reported that included both a hypervirulence and a carbapenemase-encoding plasmid [5]. The optimal treatment of infections due to the carbapenemase-producing *K. pneumoniae* remains unknown [6]. Colistin is one of the antibiotics of last resort for the treatment of such infections, but *K. pneumoniae* isolates resistant to this drug have already been reported [7], and more than 10% of the isolates are estimated to be colistin-resistant in Greece [8]. Five such isolates are described in the present study. 

The development of novel antimicrobial drugs and therapies is hindered by the fact that many resistance mechanisms have not yet been revealed, although the increasing availability of whole-genome sequencing (WGS) greatly facilitates such investigations. Thus, additional data linking antimicrobial resistance determined by traditional methods with genomic traits are required. A number of studies comparing *K. pneumoniae* resistomes determined by traditional and in silico methods were conducted [4,9,10,11,12], some of which were based on long-read WGS using the Oxford Nanopore MinION sequencing system [9,10]. However, such a comparison would be much more useful if supplemented by broad epidemiological surveillance data, including the implementation of various molecular typing schemes, and by a thorough investigation of the associated virulence factors and plasmid resistance determinants.

In this paper, we provide the comprehensive data for 36 clinical *K. pneumoniae* isolates belonging to various sequence types (ST), including two novel ones not described before. Among these isolates, 32 were multidrug-resistant (MDR) according to the commonly used definition [13]. We provide the results of isolate molecular typing using several schemes (MLST, capsular and oligosaccharide typing). 

The MLST scheme for *K. pneumoniae* developed in 2005 [14] includes the determination of internal fragments for the following seven housekeeping genes: *gapA* (glyceraldehyde 3-phosphate dehydrogenase), *infB* (translation initiation factor 2), *mdh* (malate dehydrogenase), *pgi* (phosphoglucose isomerase), *phoE* (phosphorine E), *rpoB* (beta-subunit of RNA polymerase) and *tonB* (periplasmic energy transducer), and a comparison of their allelic profiles with the ones from public databases [15]. Currently, more than 4000 sequence type profiles for *K. pneumoniae* are contained in the Institut Pasteur MLST system [15].

Additional isolate typing schemes for *K. pneumoniae* included the ones based on capsular (K, or KL) loci and oligosaccharide (O) loci. The current versions of the corresponding databases contain about 100 KL-type profiles and 12 O-profiles [16]. KL1–KL77 correspond to the loci associated with each of the 77 serologically defined K-type references, and KL101 and above are defined from the DNA sequence data on the basis of the gene content.

There are many other available typing schemes, for example, based on sequence properties [17], but in this study we used a combined approach including three molecular-based schemes.

The data also comprises antimicrobial resistance information, including plasmid determination, virulence genes’ presence and several typing schemes. The resistance/susceptibility data were acquired by both traditional methods and bioinformatics-based methods for the bacterial whole genomes obtained by second and third generation (long-read) sequencing.

## 2. Results

### 2.1. Typing and Classification

In Table 1, the results of typing our isolates using the three schemes described above are presented. The isolates belong to 16 different sequence types, with ST395 being the most abundant (seven isolates). In addition, there are 18 different KL-types and 9 O-types. Four isolates belong to the KL2 type, which is usually considered to be associated with high virulence [18]. The most frequent O-type was O2v2 (11 isolates in total).

Three isolates (P-45, P-212 and P-259) possessed novel combinations of MLST-gene alleles, i.e., a new sequence type (ST), between which P-212 and P-259 had the same ST. The last two samples were taken from the same patient, but from different sites and in different times to ensure the presence of a novel ST. We have submitted these isolates to the Institut Pasteur MLST system [15], and new MLST profiles were created for them: ST4060 for P-45 and ST4061 for the two other isolates. The allelic profiles of these new sequence types are presented in Table 2.

In order to obtain additional information regarding bacteria similarity, a maximum likelihood tree was built based on the core genome of 36 isolates containing 3888 genes. The tree is presented below in Figure 1.

It is worth noting that the KL-type provides additional classification narrowing in comparison with the sequence type alone. For example, P-183 forms a standalone clade since it possesses the KL-type (KL23) different from its neighbors’ type (KL24), although they share the same ST (ST11). Thus, such a hybrid typing scheme could be applied for a precise distinguishing of the isolates in clinical settings, which is an important task in epidemiological surveillance and infection prevention.

### 2.2. Antibiotic Resistance Determination

We have tested the isolates for antimicrobial drug resistance both by empirical (disc diffusion) and theoretical (searching for known acquired resistance genes in genomic sequences) methods. Although the empirical methods are more reliable, they undoubtedly require more time and resources. When the time is a critical factor, e.g., for patients with a severe unidentified bacterial infection, a quick bioinformatics scan for resistance based on a whole-genome sequence may become a promising option for future studies using third generation sequencers like MinION. However, currently, this technology cannot be readily implemented in routine clinical diagnostics since it allows only detecting the presence of some gene, but cannot reveal its expression or activation state. 

Thus, in order to facilitate the development of such novel point-of-care resistance determination systems, a comprehensive comparison of empirical and theoretical susceptibility determination is required. We have performed such a comparison for all 36 *K. pneumoniae* isolates. The data obtained are summarized in Figure 2.

Phenotype studies have shown that 32 of the 36 isolates were MDR. Three isolates (P-28, P-29, and P-237) were resistant to 12 antimicrobial drugs that have been tested for all the isolates. Additional studies have shown that P-28, P-29 and P-237 were resistant to many other drugs, including commercially available *Klebsiella* bacteriophages, and were susceptible only to moxifloxacin (P-29, partially) and nalidixic acid (P-237, partially). P-28 was resistant to all antimicrobials from the extended panel.

Five isolates, including P-28, P-29 and P-237 mentioned above, were colistin-resistant, which is rather rare for *K. pneumoniae* [7], since colistin is one of the antibiotics of last resort for the treatment of carbapenem-resistant *K. pneumoniae* infections [7]. Interestingly, all three isolates of ST377 (P-26, P-28, P-152) were MDR. According to the isolates’ data available in the Institut Pasteur database [15], this sequence type is rather rare (only eight fully sequenced isolates in the Pasteur database, and only one in Genbank), and is usually attributed to MDR or extensively drug-resistant (XDR) bacteria. Thus, it requires specific attention in future studies.

All isolates were ampicillin-resistant, which corresponds well with their possession of SHV-type beta-lactamase genes revealed by bioinformatics methods. These genes are known to be chromosome-encoded and provide ampicillin resistance for *K. pneumoniae* [19]. Most of the isolates were also resistant to other beta-lactam antibiotics, and half of the isolates (18/36) were resistant to carbapenems. This fact complies with the presence of the beta-lactamase genes *bla*_CTX-M-15_ (21 isolates), *bla*_TEM-1B_ (20 isolates), *bla*_OXA-1_ and *bla*_OXA-48_ (15 and 12 isolates, respectively). In addition, 26 isolates were resistant to ciprofloxacin, and this resistance is reflected in the presence of the *aac(6′)-lb-cr* gene, which is known to provide fluoroquinolone resistance in *Enterobacteriaceae* [20].

Despite the fact that the antimicrobial resistance phenotypes and genotypes correlated in general, some discrepancies were found. This is not surprising because the presence of a resistance gene in a genome cannot be considered as sole evidence for its expression and activity in the given isolate. In addition, colistin resistance genes were not revealed by bioinformatics methods. However, ColRNAI plasmids known to carry such genes [21] were found (see below). Difficulties in accurately identifying colistin resistance in vitro may also be at play with this discrepancy.

### 2.3. Plasmids

Since plasmids play a key role in acquired antimicrobial drug resistance, it is important to reveal the plasmid sequences in the WGS data. This information will greatly facilitate the investigations of resistance mechanisms. We have performed an in silico search for plasmids in all genomic sequences of the isolates. In addition, sequencing with long reads usually provides necessary resolution to separate genomic and plasmid sequences during reads’ assembly to contigs [22], so we have also checked the plasmid presence in hybrid short–long reads’ assemblies for the 14 selected isolates. The list of plasmids identified in the isolates is shown below in Table 3.

The plasmid data correspond with the antimicrobial resistance data obtained by the traditional and in silico methods. For example, IncF-type plasmids revealed in most isolates are commonly reported worldwide to carry carbapenemase genes and quinolone resistance genes [23], which were found in most isolates. 

The ColRNAI plasmid carrying colistin resistance genes was revealed in three MDR (P-28, P-29 and P-152) isolates belonging to the ST377 sequence type. This plasmid was also found in hypervirulent P-108 and P-140, but their phenotypic resistance/sensitivity to colistin was not confirmed, as well as for P-152, since we have not performed the serial dilution approach to confirm their sensitivity. Nevertheless, the isolates having this plasmid require additional attention. At the same time, the other colistin-resistant isolates P-26 and P-200 had an IncH plasmid that is associated with the *mcr-1* and *mcr-3* plasmid-mediated colistin resistance genes in *E. coli* [23,24]. However, *mcr* genes have not been revealed in P-26, P-28, P-29 or P-200, which suggests some different resistance mechanism. In P-237, the plasmid has not been revealed either. However, the long-read sequencing of the selected samples confirmed the possibility of the plasmid mechanism of antimicrobial resistance acquisition to other antibiotics in most isolates.

In general, although the long-read sequencing allowed to obtain the reliable whole-plasmid sequences, it did not provide additional information regarding plasmid typing. Deeper investigations of plasmid sequences and structure are required to get additional insights in resistance transfer mechanisms, but such studies lie beyond the scope of this manuscript.

### 2.4. Virulence Genes

Besides the antibiotic resistance data, another important piece of information is the virulence of the investigated isolates that represents the degree of damage caused by a bacterium to its host. Although the bacteria of some sequence types, especially ST23, were considered to be hypervirulent [3], it is not always possible to link the sequence type to the virulence characteristics which contribute to the pathogenicity of the bacteria. Thus, it would be useful to identify known virulence factor genes in the genomes of the isolates investigated by bioinformatics methods.

*K. pneumoniae* has many virulence factors, for example, lipopolysaccharide, fimbriae, outer membrane proteins and determinants of iron acquisition and nitrogen source utilization [25].

Several panels of genes encoding the virulence factors were proposed. Usually, the genes specifying the K1 (*magA*) and K2 (*wzi*) serotypes, together with seven other virulence factors (*rmpA*, *entB*, *ybtS*, *kfu*, *iutA*, *mrkD*, *allS*) are considered important [18,26]. The less studied factors associated with *K. pneumoniae* virulence include outer membrane proteins (porins) and the allantoin metabolism system [1].

The complete list of detected virulence genes is shown in the Appendix A, and the presence of the main virulence factors given above is shown in Figure 3.

All isolates possessed *allS* (allantoinase gene) and *entB* (enterobactin), while *mrkD* (fimbriae, 28 isolates) and *ybtS* (yersiniabactin, 18 isolates) were rather common as well. In addition, most isolates (from 26 to 33 for different genes) contained other genes of the *mrk* cluster, which are involved in type 3 fimbriae formation [1], suggesting that this virulence mechanism is likely to be important in the isolates under study. The *rmpA* (regulator of mucoid phenotype A) gene was rare and was revealed only in two isolates (P-108, P-140) belonging to the hypervirulent ST23 type. This high degree of virulence was also supported by the fact that only these two isolates possessed 6 of 7 virulence genes studied. Genes from the *kfu* cluster (iron uptake system) were revealed in four isolates, including two (P-212 and P-259) possessing the novel sequence type ST4061. The aerobactin transporter gene (*iutA*) was found in 10 isolates. A half of all isolates contained four or more virulence factors, suggesting that they could be dangerous in clinical settings, as they are also MDR, and require further investigations.

## 3. Discussion

To the best of our knowledge, this is the first detailed molecular epidemiology report based on second and third generation whole-genome sequencing for clinical isolates of *K. pneumoniae* in the Russian Federation. In this study two new MLST-based sequence types represented by one and two isolates, respectively, were revealed. Both sequence types were deposited to the MLST database of the Institut Pasteur (Paris, France) [15] as ST4060 and ST4061, respectively. We plan to collect additional samples to investigate the frequency of these sequence types and possible ways for their transmission. Since 36 isolates belonged to 16 different sequence types and 18 different KL-types without a significant prevalence of any type or spreading across all clinical departments, the data obtained cannot prove that all or most of the isolates had a nosocomial origin. Although this study included taking samples from patients visiting the medical center from various regions of the Russian Federation, it obviously cannot be considered as being representative of the whole country due to the very limited sample population. 

Our study has confirmed the negative tendency of carbapenemase-producing pathogens’ spread in various Russian high-technological medical centers described recently [27] and in earlier publications [28,29]. We believe that such investigations performed by us and our colleagues will highlight the importance of carbapenemase-producing pathogen surveillance and stimulate similar studies in other medical centers from different parts of Russia. In our opinion, currently, such investigations are scarce and their value is underestimated, but the situation gradually changes for the better.

It is also interesting that no ST258 isolates were revealed, although this sequence type is rather common in Europe, North America and Australia [30]. However, 12 members of the clonal group 258 [30] including, among others, ST11 and ST395 were found. 

Thirty-two of 36 (89%) isolates expressed an MDR phenotype, 18 of them (50%) were resistant to carbapenems, and 5 to colistin (14%), while 3 were resistant to the whole panel of 12 antibiotics belonging to various classes. The in silico screening for the *mcr* genes in colistin-resistant isolates did not reveal any results, but the P-29 isolate carried a C->A mutation in position 29 of the *mgrB* gene, leading to a C39F amino acid substitution, which, in turn, could lead to function deactivation according to the PROVEAN server (http://provean.jcvi.org/index.php). Other isolates exhibited a wild-type *mgrB*, so the mechanisms of their resistance remain to be elucidated. This partially complies with the results obtained in recent colistin resistance investigations of *K. pneumoniae* from Greece [31]. In addition, two isolates were hypervirulent that was confirmed by the presence of six important virulence factors in their genomes and their belonging to the ST23 sequence type. All three isolates attributed to the rarely revealed ST377 were MDR, including the one resistant to the extended panel, so this sequence type should be extensively studied in the future.

In most cases, there was a good compliance between the phenotypic and genotypic antibiotic resistance, but some discrepancies were revealed, in particular, for colistin, which suggests a possible resistance mechanism based on the chromosomal gene inactivation in such cases.

Long-read sequencing on MinION (Oxford Nanopore Technologies, Oxford, UK) allowed to greatly improve the assembly of 14 *K. pneumoniae* genomes (down to one contig for each genome), including the distinguishing of the plasmid sequences. The data obtained will facilitate future studies of antimicrobial resistance mechanisms in *Enterobacteria*.

In general, the prevalence of antibiotic resistance in the isolates with the hypermucoviscosity phenotype (hvKP, highly virulent) is rare compared with the high prevalence of antibiotic-resistant isolates [1,32]. However, with the global dissemination of mobile genetic elements harboring various antibiotic resistance genes, including the *K. pneumoniae* carbapenemase (KPC), New Delhi metallo-beta-lactamase (NDM) and oxacillinases-48 (OXA-48) types of carbapenemases, antibiotic-resistant hvKP isolates have begun to emerge in recent years [32]. In our study, we have determined two highly virulent isolates belonging to ST23. Although they had an MDR phenotype, their resistance, on average, was lower than for the less virulent isolates. In addition, the presence of five virulence factors in the highly resistant P-152 isolate belonging to ST377 need further investigation due to the possible emergence of a novel dangerous source of infection.

## 4. Materials and Methods

### 4.1. DNA Isolation, Sequencing and Genome Assembly

Thirty-six patient samples were obtained from 34 patients (21 males and 13 females) in various sources and clinical departments (see Table 1) of a multipurpose medical center in Moscow, Russia, during the period 2017–2018. Patients’ ages ranged from 24 to 98 with a median equal to 60 years. At the first step, 32 isolates resistant to at least 3 antimicrobial agents, as defined by the phenotype, were randomly selected from the set of available isolates. Then four additional isolates (P-6, P-15, P-45 and P-190) that have caused severe infections were included, even though they did not exhibit an MDR phenotype, in order to better investigate these cases and to exclude possible determination errors. The genomic DNA was isolated with a DNeasy Blood and Tissue kit (Qiagen, Hilden, Germany) and used for the paired-end library preparation with a Nextera^™^ DNA Sample Prep Kit (Illumina^®^, San Diego, CA, USA), and the WGS of all 36 isolates was performed on Illumina^®^ Hiseq or MiSeq platforms. 

The WGS was also performed using the Oxford Nanopore MinION sequencing system (Oxford Nanopore Technologies, Oxford, UK). DNA was used to prepare the MinION library with the Rapid Barcoding Sequencing kit SQK-RBK004 (Oxford Nanopore Technologies, Oxford, UK). The amount of initial DNA used for the barcoding kit was 400 ng for each sample. All mixing steps for the DNA samples were performed by gently flicking the microfuge tube instead of pipetting. All libraries were prepared according to the manufacturer’s protocols. The final library was sequenced on an R9 SpotON flow cell. The standard 24 h sequencing protocol was initiated using the MinKNOW software (Oxford Nanopore Technologies, Oxford, UK). Fourteen of 36 isolates considered interesting according to various criteria were sequenced.

Base calling of the raw MinION data was performed with the Guppy Basecalling Software version 2.3.1 (Oxford Nanopore Technologies, Oxford, UK). Demultiplexing was made using Deepbinner version 0.2.0 [33]. Assemblies were obtained using SPAdes version 3.10 [34] (Illumina sequencing), Canu version 1.8 [35] (MinION sequencing) and Unicycler version 0.4.8-beta [36] (hybrid assemblies). 

The statistics showing the parameters of the isolate sequencing and assembly are shown in Table 4. All assemblies were uploaded to NCBI Genbank under the project number PRJNA580263.

### 4.2. Ethical Statement

Ethical approval was not required as human samples were routinely collected and patients’ data remained anonymous. The planning conduct and reporting of the study was in line with the Declaration of Helsinki, as revised in 2013.

### 4.3. Determination of Antibiotic Susceptibility

All isolates were identified down to a species level by time-of-flight mass spectrometry (MALDI-TOF MS) using the VITEC MS system (bioMerieux, Marcy-l’Étoile, France). The susceptibility was determined by the disc diffusion method using the Mueller–Hinton medium (bioMerieux, Marcy-l’Étoile, France) and disks with antibiotics (BioRad, Marnes-la-Coquette, France), and by the boundary concentration method on a VITEK2Compact30 analyzer (bioMerieux, Marcy-l’Étoile, France). The isolates were tested for susceptibility/resistance to the following drugs: amikacin, ampicillin, aztreonam, cefepime, cefotaxime, ceftazidime, ceftriaxone, ciprofloxacin, colistin, doripenem, gentamicin and tigecycline. Imipenem and meropenem were also tested, but yielded the same results as doripenem in terms of resistance/susceptibility, so they are not included in the output results. The panel of antimicrobial compounds included for testing in this study reflected those agents used for human therapy in the Russian Federation. To interpret the results obtained, we used the clinical guidelines “Determination of the susceptibility of microorganisms to antimicrobial drugs”, version 2015-02 (http://www.antibiotic.ru/minzdrav/files/docs/clrec-dsma2015.pdf) which are based on EUCAST 2015.

### 4.4. Data Processing

The assembled genomes were processed using a custom software pipeline including a set of scripts for the seamless integration of various available software tools. The main goal of the investigations was to determine the antibiotic resistance in silico, including both of the known acquired resistance genes and mutations facilitating the development of such a resistance. The parameters useful for epidemiological surveillance and investigating the presence of virulence factors were also studied. 

The first step of the pipeline includes running BLAST on the “nt” database from NCBI locally to verify that the genomic sequence really belongs to the organism of interest. Then, multilocus sequence typing (MLST) [37] was performed. After that, ResFinder [38] was called to check for the presence of known antibiotic-resistant genes in the genomic sequence. Next, the steps included searching for known virulence genes with VirulenceFinder [39] and detecting the plasmids in the input sequence using PlasmidFinder [40]. In addition, it was reported that some infection control strategies involve targeting the *Klebsiella* surface polysaccharides [16], which are engaged in the virulence mechanisms. Therefore, we also used Kaptive [16] to search for the K- and O-loci of *K. pneumoniae,* which allow for defining the capsule synthesis type and lipopolysaccharide serotype, respectively, thus narrowing the isolate classification. Finally, the results obtained by the programs mentioned above were summarized and presented in a concise tabular form.

To build the phylogenetic tree representing the relations between the isolates, Prokka [41] was used for the gene annotation, Roary [42] for obtaining the core genome and RAxML [43] for building the maximum likelihood tree itself. The graphs representing the antibiotic resistance and virulence factor presence were built using the ggplot2 R package.

## 5. Conclusions

In conclusion, here we presented a general report containing various types of molecular-based typing, antibiotic resistance profiling, virulence factors and plasmid descriptions for 36 clinical *K. pneumoniae* isolates from Moscow, Russia, among which 32 isolates exhibited an MDR phenotype. Although WGS in general, and long-read sequencing in particular, allows getting some insights into the resistance transmission and spreading of pathogenic isolates across hospital departments, more research, which is currently underway, is needed to analyze the mechanisms of the resistance acquisition and virulence of these isolates.

## Figures and Tables

**Figure 1 antibiotics-09-00261-f001:**
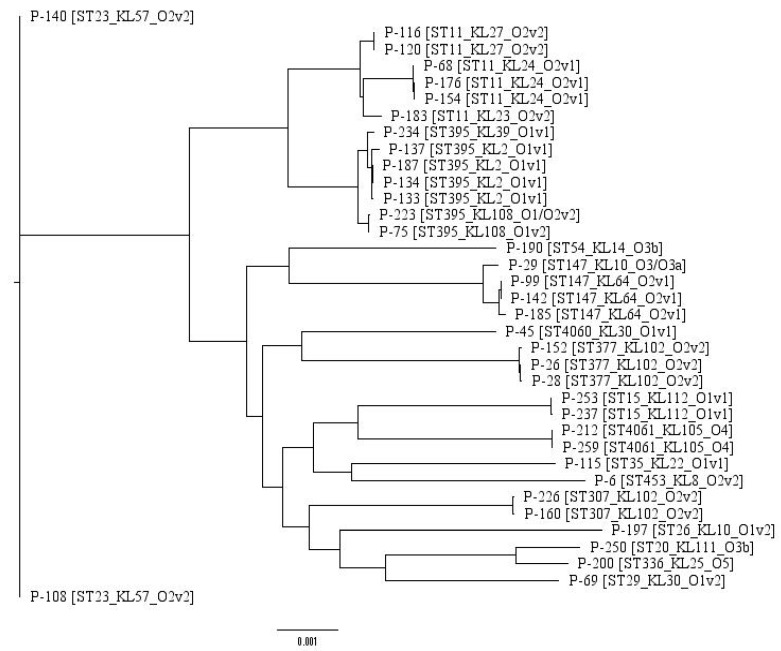
Maximum likelihood phylogenetic tree of the 36 *K. pneumoniae* isolates studied.

**Figure 2 antibiotics-09-00261-f002:**
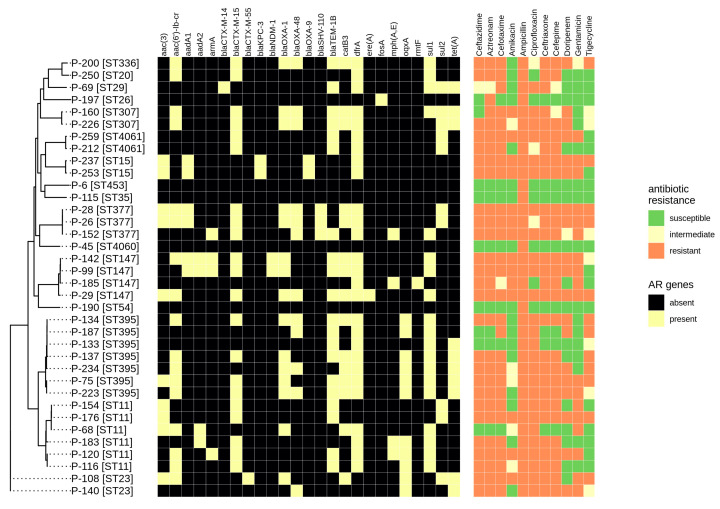
Phenotypic and genotypic antibiotic resistance profiles of the isolates studied. Results for colistin phenotype are described in the text.

**Figure 3 antibiotics-09-00261-f003:**
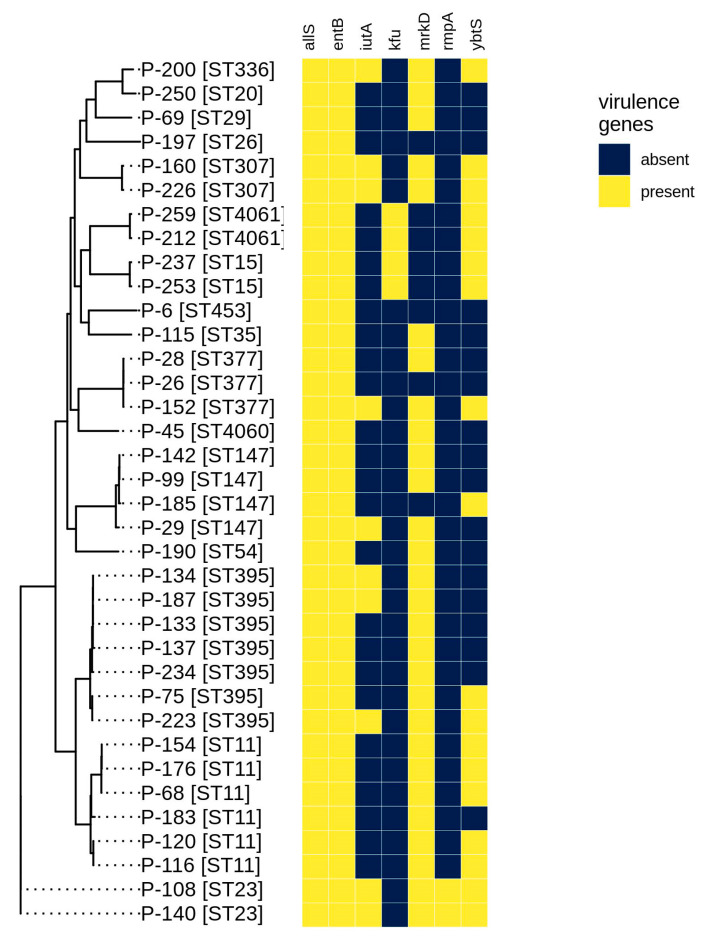
Virulence gene presence for the isolates studied revealed by bioinformatics methods.

**Table 1 antibiotics-09-00261-t001:** Description and molecular typing (multilocus sequence typing (MLST) (ST), capsule synthesis (KL)-type and lipopolysaccharide serotype (O-type)) of the isolates used in the study.

Sample id	Patient Code	Year Collected	Hospital Department	Site	ST	KL-Type	O-Type
**P-6**	A1	2017	ICU	blood	ST453	KL8	O2v2
**P-26**	A2	2017	Surgery	traumatic discharge	ST377	KL102	O2v2
**P-28**	A3	2017	ICU	tracheal aspirate	ST377	KL102	O2v2
**P-29**	A4	2017	ICU	tracheal aspirate	ST147	KL10	O3/O3a
**P-45**	A5	2017	Urology	Sputum	**ST4060***	KL30	O1v1
**P-68**	A6	2017	Rehabilitation	Urine	ST11	KL24	O2v1
**P-69**	A7	2017	ICU	Sputum	ST29	KL30	O1v2
**P-75**	A8	2017	ICU	Blood	ST395	KL108	O1v2
**P-99**	A9	2017	ICU	Urine	ST147	KL64	O2v1
**P-108**	A10	2017	Thoracic	tracheal aspirate	ST23	KL57	O2v2
**P-115**	A11	2017	Rehabilitation	Urine	ST35	KL22	O1v1
**P-116**	A12	2017	ICU	Blood	ST11	KL27	O2v2
**P-120**	A12	2017	ICU	granulation tissue	ST11	KL27	O2v2
**P-133**	A13	2017	Rehabilitation	Urine	ST395	KL2	O1v1
**P-134**	A14	2017	Hematology	Blood	ST395	KL2	O1v1
**P-137**	A15	2017	Gastroenterology	Urine	ST395	KL2	O1v1
**P-140**	A16	2017	Hematology	Blood	ST23	KL57	O2v2
**P-142**	A17	2017	ICU	Faeces	ST147	KL64	O2v1
**P-152**	A18	2017	Urology	Urine	ST377	KL102	O2v2
**P-154**	A19	2017	ICU	Blood	ST11	KL24	O2v1
**P-160**	A20	2017	Hematology	nasal swab	ST307	KL102	O2v2
**P-176**	A21	2017	ICU	Faeces	ST11	KL24	O2v1
**P-183**	A22	2017	Cardiology	Urine	ST11	KL23	O2v2
**P-185**	A23	2017	Rehabilitation	Urine	ST147	KL64	O2v1
**P-187**	A24	2017	Surgery	traumatic discharge	ST395	KL2	O1v1
**P-190**	A25	2017	Therapy	Sputum	ST54	KL14	O3b
**P-197**	A26	2018	ICU	Gall bladder	ST26	KL10	O1v2
**P-200**	A27	2018	Cardiology	Sputum	ST336	KL25	O5
**P-212**	A28	2018	ICU	bronchoalveolar lavage	**ST4061**	KL105	O4
**P-223**	A29	2018	ICU	traumatic discharge	ST395	KL108	O1/O2v2
**P-226**	A30	2018	Surgery	Blood	ST307	KL102	O2v2
**P-234**	A31	2018	ICU	abdominal cavity apostem	ST395	KL39	O1v1
**P-237**	A32	2018	ICU	Blood	ST15	KL112	O1v1
**P-250**	A33	2018	Hematology	Blood	ST20	KL111	O3b
**P-253**	A34	2018	ICU	Urine	ST15	KL112	O1v1
**P-259**	A28	2018	ICU	Faeces	**ST4061**	KL105	O4

*novel sequence types found by us are shown in bold; the isolates sequenced on MinION are filled with a gray color; ICU–intensive care unit.

**Table 2 antibiotics-09-00261-t002:** Allelic profiles for novel *K. pneumoniae* sequence types ST4060 and ST4061.

*gapA*	*infB*	*mdh*	*pgi*	*phoE*	*rpoB*	*tonB*	ST
2	4	2	1	26	1	2	4060
4	1	1	26	66	4	472	4061

**Table 3 antibiotics-09-00261-t003:** Plasmid sequences found in the isolates.

Sample id\Plasmid Type	Col	Inc (Other)	IncR
P-6	Col(BS512)	IncFIB(K)_Kpn3	-
P-26	ColRNAI	IncHI1B_pNDM-MAR, IncL/M(pOXA-48)_pOXA-48, IncQ1	-
P-28	ColRNAI	IncHI1B_pNDM-MAR, IncL/M(pOXA-48)_pOXA-48	-
P-29	Col(BS512)	-	IncR
P-45	-	-	-
P-68	Col(BS512)	IncFIB(K)_Kpn3, IncFII(K)	IncR
P-69	-	-	-
P-75	Col(BS512)	IncFII	IncR
P-99	-	IncFIB(pQil)_pQil, IncHI1B_pNDM-MAR	-
P-108	ColRNAI	IncFII, IncQ1	-
P-115	-	-	-
P-116	-	IncL/M	IncR
P-120	-	IncL/M	IncR
P-133	-	-	IncR
P-134	-	IncL/M(pOXA-48)_pOXA-48, IncX4	IncR
P-137	-	IncFIB(Mar)_pNDM-Mar	IncR
P-140	ColRNAI	IncL/M(pOXA-48)_pOXA-48	-
P-142	-	IncFIB(pQil)_pQil, IncHI1B_pNDM-MAR	-
P-152	ColRNAI	IncHI1B_pNDM-MAR, IncL/M	-
P-154	Col(BS512)	-	IncR
P-160	-	-	-
P-176	Col(BS512)	-	IncR
P-183	ColRNAI	IncA/C2, IncFIB(K)_Kpn3, IncFII(K)	-
P-185	Col440II	IncFII(pKPX1)_AP012055, IncL/M(pOXA-48)_pOXA-48	IncR
P-187	-	IncL/M	IncR
P-190	-	-	-
P-197	-	-	IncR
P-200	-	IncHI1B_pNDM-MAR	-
P-212	-	IncFII(K)	-
P-223	-	IncL/M(pOXA-48)_pOXA-48	IncR
P-226	-	IncL/M(pOXA-48)_pOXA-48	-
P-234	Col(BS512)	IncL/M(pOXA-48)_pOXA-48	IncR
P-237	Col(BS512)	IncFIB(pQil)_pQil	-
P-250	-	-	-
P-253	Col(BS512)	IncFIB(pQil)_pQil	-
P-259	-	IncFII(K)	-

**Table 4 antibiotics-09-00261-t004:** Sequencing and assembly characteristics for the 36 isolates studied.

Id	Num_Reads	Num_Contigs	Coverage	N50
**P-6**	9132788	347	120	141817
**P-26**	7655908	94	87	225016
**P-28**	11221516	91	124	249107
**P-29**	8340716	121	83	319263
**P-45**	11426428	52	192	316824
**P-68**	14543708	67	183	261596
**P-69**	15629276	57	210	407523
**P-75**	2399707	164	64	145594
**P-99**	9478824	101	105	206727
**P-108**	7790180	82	133	307643
**P-115**	4387439	102	120	174171
**P-116**	4865862	114	136	159297
**P-120**	4977489	155	138	157758
**P-133**	6947364	78	125	216945
**P-134**	6276120	126	101	160646
**P-137**	7006012	103	101	195276
**P-140**	8749020	85	144	200903
**P-142**	9410400	89	162	277906
**P-152**	577309	132	28	164261
**P-154**	496548	92	26	163095
**P-160**	567724	134	28	203932
**P-176**	17679448	78	200	262658
**P-183**	2441972	128	50	179904
**P-185**	2799220	92	53	230734
**P-187**	3693052	139	75	134707
**P-190**	2891576	157	59	177583
**P-197**	18464812	156	214	145689
**P-200**	22009116	114	248	264170
**P-212**	7357356	71	155	353295
**P-223**	11821000	114	222	260983
**P-226**	9195800	107	175	280687
**P-234**	9399924	95	155	207047
**P-237**	4088672	82	87	209627
**P-250**	18130280	97	209	345165
**P-253**	16252888	77	173	196726
**P-259**	5654368	108	64	241983

The isolates marked in gray color were also sequenced on MinION.

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
