# Peer review of "Molecular Typing, Characterization of Antimicrobial Resistance, Virulence Profiling and Analysis of Whole-Genome Sequence of Clinical Klebsiella pneumoniae Isolates"

_antibiotics, 2020, doi:10.3390/antibiotics9050261_

Round 1

Reviewer 1 Report

The authors performed sequencing for clinical isolates of K. pneumoniae. They  revealed two new MLST-based sequence types. They carefully described the study design, experiment, and data processing. The conclusion is well supported by the results.

My recommendation is to accept the manuscript for publication after minor revision.
I feel the discussion is too short. The author may want explain some limitation of their study and future work (although they already described some in the manuscript) based on their novel STs.

Author Response

Reviewer 1.

My recommendation is to accept the manuscript for publication after minor revision.
I feel the discussion is too short. The author may want explain some limitation of their study and future work (although they already described some in the manuscript) based on their novel STs.

We have extended the discussion according to the given suggestions (lines 218-225, and, additionally, 226-232).

Reviewer 2 Report

Shelenkov et al investigated carbapenemase producing isolates K pneumoniae in the Russian federation. Methods were appropriate.  The advantage of the paper is the usage of WGS which gives full insight of all the mechanisms of resistance and virulence potential of the isolates.

I have some comments:

Major comments

  1. Most important is to correct the number of extremely drug resistant isolates. According to the criteria of Magiorakos, 2012 XDR is when ”the isolate is non-susceptible to at least 1 agent in all but 2 or fewer antimicrobial categories in Table 1c”. In table 1c (Magiorakos, 2012) all groups of antimicrobials are defined. From these groups the authors didn’t test antipseudomonal penicillins + inhibitors, Cephamycins, Phenicols, Phosphonic acids. The result for the second generation cephalosporins and penicillins + inhibitors should also be mentioned. The authors should either retest isolates or report only multidrug resistant isolates.
  2. The authors report that criteria for the inclusion of the strains is the resistance to three groups of antimicrobials, to which groups were non-susceptible strains P-6, -15,-190,-45. From the Fig 2 they seem to be resistant only to ampicillin. The criteria for inclusion of the strains should be clearer. Have the authors collected all isolates with MDR phenotype or have they randomly chosen?
  3. Line 274/285 How did the authors perform susceptibility testing to colistin? This should be explained separately. From line 163/164 it seems that they didn’t do MIC testing with broth microdilution  technique – if so the results for colistin were unacceptable.
  4. Table 4, Table 5 and fig 3 could be combined into one table that could include the most important ESBL/ carbapenemases genes, ST types and KL types. Now it is very difficult to comprehend and analyze the data.
  5. Why have the authors decided to use doripenem? Imipenem and meropenem are the better choice for treatment. In my opinion these antimicrobials should be tested.
  6. The authors mentioned 5 colistin resistant strains. From WGS results some additional information could be found about mechanisms of resistance – presence of mcr, status of mgr
  7. Line 109/117 Did the authors really think that next generation sequencing could be used for unidentified bacterial infections in order to facilitate the choice of antibiotic treatment and that it is faster than normal antimicrobial susceptibility testing? Moreover, detection of genes for resistance is not enough to predict antimicrobial susceptibility results as the important role has the expression of the genetic determinants. Please correct the sentence.
  8. The discussion part is a little bit short. The authors could have discussed more about detected carbapenemases - the situation with such enzymes in Russia, also they could compare more detailed virulent isolates and carbapenemase producing ones.

Minor coments

  1. The title could be shortened –“using second- and third-generation sequencing” could be deleted.
  2. Line 72/85 – this information should be moved to the introduction section.
  3. Table 3 Please move the ST number near the designation of the isolate and remove the colours showing different ST type. In the present way it is not easily understandable.

Author Response

Reviewer 2.

Thank you for thorough and useful comments that have led to significant improvement of the manuscript’s readability.

The paper has been re-written substantially, including table re-numbering and adding references, so please refer to new table and line numbers shown in the answers below.

Major comments

  1. Most important is to correct the number of extremely drug resistant isolates. According to the criteria of Magiorakos, 2012 XDR is when ”the isolate is non-susceptible to at least 1 agent in all but 2 or fewer antimicrobial categories in Table 1c”. In table 1c (Magiorakos, 2012) all groups of antimicrobials are defined. From these groups the authors didn’t test antipseudomonal penicillins + inhibitors, Cephamycins, Phenicols, Phosphonic acids. The result for the second generation cephalosporins and penicillins + inhibitors should also be mentioned. The authors should either retest isolates or report only multidrug resistant isolates. We have corrected the description to report only multidrug resistant isolates. The extended panel of antibiotics was used only for several isolates, so it is inappropriate to report MDR and XDR based on different panels.
  2. The authors report that criteria for the inclusion of the strains is the resistance to three groups of antimicrobials, to which groups were non-susceptible strains P-6, -15,-190,-45. From the Fig 2 they seem to be resistant only to ampicillin. The criteria for inclusion of the strains should be clearer. Have the authors collected all isolates with MDR phenotype or have they randomly chosen? The isolates with MDR phenotypes were randomly chosen, and four isolates mentioned were added since they have caused severe sepsis, in order to investigate these cases further and check for possible determination or isolation errors. These was made clear in lines 269-273.
  3. Line 274/285 How did the authors perform susceptibility testing to colistin? This should be explained separately. From line 163/164 it seems that they didn’t do MIC testing with broth microdilution  technique – if so the results for colistin were unacceptable. We have not performed MIC testing with broth microdilution technique, so colistin susceptibility results cannot be confirmed, and we removed all the statements regarding colistin susceptibility from the manuscript. However, according to the clinical guidelines used ( “Determination of the susceptibility of microorganisms to antimicrobial drugs”, version 2015-02 (http://www.antibiotic.ru/minzdrav/files/docs/clrec-dsma2015.pdf)) which are based on EUCAST 2015, if Vitek2Compact outputs the “Resistant” result for colistin, it can be considered appropriate.
  4. Table 4, Table 5 and fig 3 could be combined into one table that could include the most important ESBL/ carbapenemases genes, ST types and KL types. Now it is very difficult to comprehend and analyze the data. Tables 4 includes plasmids revealed, and table 5 contains virulence genes, while figure 3 also depicts virulence genes, and not resistance genes. The resistance genes and phenotypes are shown in the figure 2. So, probably, this note was made regarding both virulence and resistance genes’ representation. We have removed some genes from the figure 2, as well as sequence type coloring, to make the figure easier to understand. We has also moved the table with virulence genes to Supplementary materials (table S1), since this information could be useful for some researchers, but for most readers the data shown in the fig. 3 would, probably, be sufficient.
  5. Why have the authors decided to use doripenem? Imipenem and meropenem are the better choice for treatment. In my opinion these antimicrobials should be tested. We have tested doripenem, imipenem and meropenem for all 36 isolates. The results were exactly the same in terms of resistance/susceptibility. The actual antibiotic used for patient treatment varied among patients due to various reasons including the availability at the moment of treatment. To make this clear, we have added the statement in “Determination of antibiotic susceptibility” subsection of Materials and Methods (lines 305-307).
  6. The authors mentioned 5 colistin resistant strains. From WGS results some additional information could be found about mechanisms of resistance – presence of mcr, status of mgrThe description was added to the Discussion section (lines 238-243). Briefly, mcr genes were not found, but mgrB carried an inactivating mutation
  7. Line 109/117 Did the authors really think that next generation sequencing could be used for unidentified bacterial infections in order to facilitate the choice of antibiotic treatment and that it is faster than normal antimicrobial susceptibility testing? Moreover, detection of genes for resistance is not enough to predict antimicrobial susceptibility results as the important role has the expression of the genetic determinants. Please correct the sentence. We have described just a possible perspective, not a currently available clinical option. NGS approach can facilitate ruling out some antibiotics, but, of course, it can be unnecessary strict in this regard. We believe that the studies of phenotype-genotype match, like our study, may facilitate the future developments and investigations in this field. We have corrected the paragraph to make this clear (lines 113-120).
  8. The discussion part is a little bit short. The authors could have discussed more about detected carbapenemases - the situation with such enzymes in Russia, also they could compare more detailed virulent isolates and carbapenemase producing ones. We have extended the discussion according to the given suggestions (lines 218-232, 238-244, and 260-264). Four references have been added.

Minor coments

9. The title could be shortened –“using second- and third-generation sequencing” could be deleted. The title has been shortened according to suggestion

10. Line 72/85 – this information should be moved to the introduction section. The description of typing schemes has been moved to Introduction section (lines 65-79).

11. Table 3 Please move the ST number near the designation of the isolate and remove the colours showing different ST type. In the present way it is not easily understandable. There are neither ST numbers nor colours in Table 3 (which is now table 2 because of text rearrangement). We did not get the point. Probably, this comment addresses the figs. 2 and 3 – we have modified it to remove ST colours.

Round 2

Reviewer 2 Report

The authors  have answered in satisfactory way my questions.

Author Response

The reviewer has not provided additional comments and stated that the answers provided previously were satisfactory.